# Association between Body Mass Index and Renal Outcomes Modified by Chronic Kidney Disease and Anemia: The Obesity Paradox for Renal Outcomes

**DOI:** 10.3390/jcm11102787

**Published:** 2022-05-15

**Authors:** Chi-Chih Hung, Pei-Hua Yu, Sheng-Wen Niu, I-Ching Kuo, Jia-Jung Lee, Feng-Ching Shen, Jer-Ming Chang, Shang-Jyh Hwang

**Affiliations:** 1Division of Nephrology, Department of Internal Medicine, Kaohsiung Medical University Hospital, Kaohsiung Medical University, Kaohsiung 80708, Taiwan; chichi@kmu.edu.tw (C.-C.H.); rubyvian@gmail.com (P.-H.Y.); jjlee@kmu.edu.tw (J.-J.L.); jemich@kmu.edu.tw (J.-M.C.); sjhwang@kmu.edu.tw (S.-J.H.); 2Regenerative Medicine and Cell Therapy Research Center, Kaohsiung Medical University, Kaohsiung 80708, Taiwan; 3Graduate Institute of Clinical Medicine, College of Medicine, Kaohsiung Medical University, Kaohsiung 80708, Taiwan; 4Department of Internal Medicine, Kaohsiung Municipal Ta-Tung Hospital, Kaohsiung Medical University, Kaohsiung 80145, Taiwan; 950138kmuh@gmail.com (S.-W.N.); peyto26@hotmail.com (I.-C.K.); 5Faculty of Renal Care, College of Medicine, Kaohsiung Medical University, Kaohsiung 80708, Taiwan

**Keywords:** obesity paradox, renal outcome, chronic kidney disease, body mass index, central obesity, waist-to-hip ratio, anemia

## Abstract

Obesity-related nephropathy is associated with renal function progression. However, some studies have associated a high body mass index (BMI) with improved renal outcomes—this is referred to as the obesity paradox for renal outcomes, especially in relation to advanced chronic kidney disease (CKD). Central obesity can explain the obesity paradox in all-cause mortality. However, whether obesity or central obesity is associated with renal outcomes (renal replacement therapy or a 50% decline in the estimated glomerular filtration rate) in patients with advanced CKD remains unclear. Our study included 3605 Asian patients with CKD stages 1–5 divided into six groups according to their BMI (between 15 and 35 kg/m^2^). Through linear regression, BMI was positively associated with hemoglobin and albumin at CKD stages 4 and 5. In the competing risk Cox regression model, a high BMI (27.5–35 kg/m^2^) was associated with renal outcomes at CKD stages 1–3, but not stages 4 and 5. A high BMI was associated with renal outcomes in patients with hemoglobin ≥11 g/dL, but not <11 g/dL. A high waist-to-hip ratio was not associated with renal outcomes. We conclude that the CKD stage and anemia may explain the obesity paradox in renal outcomes in patients with CKD.

## 1. Introduction

The prevalence of a high body mass index (BMI), including being overweight and obese, is increasing globally [1] and contributes considerably to increased all-cause mortality in the general population [2]. However, being underweight is also associated with an increased risk of death, especially in Asian populations [3], and the BMI exhibits a U-shaped association with all-cause mortality in Asian populations [4]. Chronic kidney disease (CKD) is a global health problem associated with increased risks of cardiovascular morbidity and mortality, and its estimated prevalence worldwide is 11–13% [5]. In patients undergoing dialysis [6] and who have CKD [7], a high BMI is paradoxically linked with lower mortality, known as reverse epidemiology, or the obesity paradox [8]. The obesity paradox for all-cause mortality might be explained by the high prevalence of malnutrition in these patients.

Obesity causes glomerular diseases, usually referred to as obesity-related glomerulopathy [9]. Studies have revealed that obesity is a risk factor for the incidence of CKD [10,11] and end-stage renal disease (ESRD) [12]. The risk of ESRD begins to increase when the BMI exceeds 25 kg/m^2^ in the general population [12] and exceeds 35 kg/m^2^ in patients with CKD [13]. However, increasing evidence demonstrates that a high BMI is protective against renal function deterioration. Babayev et al. [14] and De Nicola et al. [15] failed to observe statistically significant higher incidence of ESRD as BMI increased, and Chang et al. [16] revealed a lower risk of renal function progression in patients with a high BMI. Huan et al. [17] revealed that a high BMI was protective of renal function deterioration in CKD stage 3 or 4 among patients with diabetes. This may be recognized as the obesity paradox for renal outcomes in the CKD population.

The obesity paradox for renal outcomes is counterintuitive. Few studies have presented this phenomenon, and even fewer have investigated the causes. Central obesity, advanced CKD, and malnutrition–inflammation may be possible explanations. Studies have demonstrated that BMI may not predict renal outcomes in patients with advanced CKD [13]. Malnutrition–inflammation is associated with protein energy wasting (PEW) [18], which progresses with CKD progression. The interplay between kidney dysfunction and systemic inflammation is delicate and readily spirals out of control due to the effects of cytokines, hormones, and uremic toxins [19]. Anemia is associated with hypoxemia and leads to kidney tubulointerstitial damage [20], contributing an additional effect on renal outcome in CKD populations. Central obesity may be a more effective indicator of mortality than BMI [21] because of the close relationship between adipose tissue dysfunction and metabolism syndrome, and our previous report revealed no central obesity paradox for mortality in patients with CKD [22].

The present study tried to find out the key to the obesity paradox for renal outcome in advanced CKD population. We hypothesized that advanced CKD and malnutrition–inflammation might modify the association between BMI and renal outcomes, and central obesity might be a more effective predictor of renal outcomes in patients with CKD. We tested these hypotheses through a cohort study including 3605 Asian patients with CKD stages 1–5 divided into groups according to their BMI or waist-to-hip ratio (WHR).

## 2. Materials and Methods

### 2.1. Study Design and Participants

This prospective observational study, the Integrated CKD Care Program in Kaohsiung for Delaying Dialysis, involving two affiliated hospitals of Kaohsiung Medical University in southern Taiwan, was conducted between 11 November 2002 and 31 May 2009, as described previously [8]. In the present study, we extended the follow-up period to 31 December 2014. The inclusion criterion was patients with CKD stages 1–5 not receiving renal replacement therapy (estimated glomerular filtration rate (eGFR) was calculated by using the Modification of Diet in Renal Disease equation). The exclusion criterion was acute kidney injury defined as a >50% decrease in eGFR within 3 months. We included 3605 patients with CKD stages 1–5. To study the impact of BMI and WHR on renal outcomes, participants were divided into six BMI groups (cutoff: 15, 20, 22.5, 25, 27.5, 30, and 35 kg/m^2^), in accordance with other Asian BMI studies [3] and WHR quintiles. Patients with an extreme BMI (<1% of participants), comprising 7 with a BMI < 14.9 kg/m^2^ and 58 with a BMI > 35.1 kg/m^2^, were excluded, as in our other study [8]. We enrolled 3605 patients with CKD stages 1–5 and a BMI of 15.0–35.0 kg/m^2^. The erythropoiesis-stimulating agent was reimbursed only in CKD stage 5. All patients provided informed consent to participate. The study protocol was approved by the Institutional Review Board of Kaohsiung Medical University Hospital. 

### 2.2. Collection of Demographic, Medical, and Laboratory Data

The baseline variables consisted of demographic features (age and sex), medical history (diabetes mellitus, hypertension, cardiovascular disease, Charlson comorbidity index score, metabolic syndrome, and malnutrition–inflammation score (MIS)), and laboratory data (eGFR, hemoglobin (Hb), albumin, C-reactive protein (CRP), total cholesterol, and triglycerides). The demographic features formed the baseline record, and the medical history was obtained through a doctor’s chart review and interview with patients. The BMI was calculated as weight in kilograms divided by the square of height in meters. Waist and hip circumferences were measured according to the World Health Organization protocol [23]. The WHR was defined as the waist circumference (cm) divided by the hip circumference (cm), and the waist-to-height ratio was defined as the waist circumference (cm) divided by height (cm). The MIS had 10 components, with severity scores from 0 (normal) to 3 (severely abnormal) in each component [24]. The five nutritional history–based components were body weight change, dietary intake, gastrointestinal symptoms, functional capacity, and comorbid conditions. We excluded dialysis vintage from the score and adopted the definition of the score for comorbid conditions from a CKD study [18]. The two physical examination components consisted of the assessment of subcutaneous body fat and signs of muscle wasting. The other three nonsubjective global assessment–based components were BMI, serum albumin level, and serum total iron-binding capacity. In a cohort of nondialyzed CKD patients, MIS > 3 offered a high sensitivity and MIS > 8 offered the best specificity for outcomes [18]. The metabolic syndrome and components included a waist circumference ≥90 cm in men or ≥80 cm in women; systolic blood pressure ≥130 mmHg or diastolic blood pressure ≥85 mmHg or hypertension; HDL cholesterol >40 mg/dL in men or >50 mg/dL in women; triglycerides ≥150 mg/dL; and fasting blood glucose ≥100 mg/dL or diabetes mellitus. The Charlson score predicts mortality in patients according to 17 comorbidities, including acute myocardial infarction, congestive heart failure, peripheral vascular disease, cerebral vascular accident, dementia, pulmonary disease, connective tissue disorder, peptic ulcer, liver disease, diabetes, diabetes complications, paraplegia, renal disease, cancer, metastatic cancer, severe liver disease, and human immunodeficiency virus [25]. The mean arterial pressure was calculated through the averaged systolic and diastolic blood pressure measured 3 months before and after enrollment, using the formula of one-third of averaged systolic blood pressure plus two-thirds of averaged diastolic pressure. Urine protein-to-creatinine ratio (Upcr) was calculated as urine protein (mg) divided by urine creatinine (g) from a random spot urine sample. Biochemistry measurements were performed during a screening visit, baseline visit, and then every 3 months, as per the protocol. The laboratory data from 3 months before baseline to 3 months after baseline were averaged and analyzed.

### 2.3. Outcomes

Renal outcomes were defined as renal replacement therapy and a 50% decline in the eGFR. All-cause mortality was ascertained by reviewing death certificates, using charts or the National Death Index. Models of all-cause mortality included patients who had undergone renal replacement therapy and were censored only at death or the end of follow-up.

### 2.4. Statistical Analysis

The baseline characteristics of all the patients were stratified according to their BMI and WHR, expressed as percentages for categorical data and recorded as mean ± standard deviation for continuous variables with an approximate normal distribution and median and interquartile range for continuous variables with skewed distribution. The differences between groups were verified by using a chi-square test for categorical variables or one-way analysis of variance for continuous variables. A Cox proportional hazards analysis was used to investigate the relationship between BMI and WHR and renal outcomes and all-cause mortality. Skewed distributed continuous variables were log transformed to attain normal distribution. Covariates were selected on the basis of clinical relevance, consistent with our previous paper [26]. The adjusted covariates included age, sex, eGFR, Upcr log, cardiovascular disease, smoking history, cancer, severe liver disease, and hypertension. The statistical analysis was performed by using SPSS for Windows software, version 20.0 (IBM, Chicago, IL, USA).

## 3. Results

### 3.1. Baseline Characteristics Relating to Body Mass Index and Chronic Kidney Disease Stage

The 3605 patients were divided into six groups according to their BMI (G1–G6), and the groups were then divided according to the patient’s CKD stage (1–3 and 4/5) (Table 1). In patients in the CKD-stage-1–3 group, age, male sex (%), and MIS ≥ 8 decreased with increasing BMI; Hb and triglyceride levels and percentage of hypertension and metabolic syndrome increased with increasing BMI. In patients in the CKD-stage-4/5 group, male sex (%) and MIS ≥ 8 decreased with an increasing BMI; the eGFR, Hb and triglyceride levels, and percentage of diabetes mellitus and metabolic syndrome increased with an increasing BMI.

### 3.2. Multivariable Linear Regression for Body Mass Index

The multivariate linear regression for the BMI (Table 2) revealed a significant relationship between BMI and diabetes mellitus, glycosylated Hb, mean blood pressure, Hb, the CRP log, and uric acid. A significant relationship between albumin and BMI was observed in patients with CKD stages 4/5.

### 3.3. Association between Body Mass Index, Renal Outcomes, and Mortality in Relation to Chronic Kidney Disease Stage

We reported the hazard ratio (HR) for renal outcomes based on BMI, using the fully adjusted Cox regression model (Table 3). For patients with CKD stages 1–3, the HR (95% confidence interval, CI) for renal outcomes was increased in the low-BMI (G1 (HR 1.58; 95% CI 0.95–2.61) and G2 (HR 1.84; 95% CI 1.27–2.66)) and high-BMI (G5 (HR 1.51; 95% CI 1.01–2.26) and G6 (HR 1.58; 95% CI 1.00–2.51)) groups compared with that of the reference BMI group (G4). For patients with CKD stages 4 and 5, the HR for renal outcomes was significantly increased only in the low BMI group (G1 (HR 1.31; 95% CI 1.07–1.61)).

We also reported the HR for all-cause mortality in terms of BMI, using the fully adjusted Cox regression model. In patients with CKD stages 1–3, the HR for all-cause mortality was increased in the low-BMI (G1 (HR 2.54; 95% CI 1.55–4.16) and G2 (HR 1.37; 95% CI 0.88–2.14)) and high-BMI (G6 (HR 1.44; 95% CI 0.86–2.42)) groups compared with that of the reference BMI group (G4). In patients with CKD stages 4 and 5, the HR for all-cause mortality was increased only in the low BMI groups (G1 (HR 1.91; 95% CI 1.30–2.80) and G2 (HR 1.58; 95% CI 1.11–2.25)) compared with that of the reference BMI group (G6).

### 3.4. Association between Body Mass Index, Renal Outcomes, and Mortality in Terms of Hemoglobin Levels

We reported the HR for renal outcomes in terms of BMI, using the fully adjusted Cox regression model (Table 4). In patients with Hb < 11 g/dL, the HR for renal outcomes was significantly increased in the low BMI groups (G1 (HR 1.48; 95% CI 1.15–1.90)) compared with that of the reference BMI group (G5). In patients with Hb ≥ 11 g/dL, the HR for renal outcomes was increased in the high BMI groups (G3 (HR 1.39; 95% CI 1.01–1.92), G5 (HR 1.79; 95% CI 1.25–2.55), and G6 (HR 1.66; 95% CI 1.14–2.43)) compared with that of the reference BMI group (G2). 

We also reported the HR for mortality in terms of BMI, using the fully adjusted Cox regression model (Table 4). In patients with Hb < 11 g/dL, the HR for mortality was significantly increased in the low-BMI group (G1 (HR 1.76; 95% CI 1.16–2.68)) compared with that of the reference BMI group (G6). In patients with Hb ≥ 11 g/dL, the HR for mortality was significantly increased in the low-BMI group (G2 (HR 2.10; 95% CI 1.19–3.71)) compared with that of the reference BMI group (G4).

### 3.5. Association between Body Mass Index and Renal Outcomes in Terms of Chronic Kidney Disease Stage

To differentiate the effect of BMI and different CKD stages, we reported the HR for renal outcomes by using the fully adjusted Cox regression model grouped by BMI. The HR for renal outcomes was significantly increased in patients with a low BMI (G1 (HR 1.38; 95% CI 1.15–1.67)) and CKD stages 1–5; significantly increased in patients with a high BMI (G6 (HR 1.35; 95% CI 1.03–1.77)) and CKD stages 1–4; and significantly increased in patients with a low BMI (G2 (HR 1.84; 95% CI 1.27–2.66)) and high BMI (G5 (HR 1.51; 95% CI 1.01–2.26)) and CKD 1–3 (Figure 1) compared with that of the reference BMI group (G4).

### 3.6. Association between Body Mass Index and Renal Outcomes According to Hemoglobin Levels

To differentiate the effect of BMI and different Hb levels, we reported the HR for renal outcomes by using the fully adjusted Cox regression model grouped by BMI. In the general population, the HR for renal outcomes was significantly increased in the low-BMI groups (G1 (HR 1.38; 95% CI 1.15–1.67)) compared with that of the reference BMI group (G4). In patients with Hb ≥ 9, the HR for renal outcomes was not significantly increased compared with the reference BMI group (G2). In patients with Hb ≥ 11 g/dL, the HR for renal outcomes was significantly increased in the high-BMI groups (G3 (HR 1.39; 95% CI 1.01–1.92), G5 (HR 1.79; 95% CI 1.25–2.55), and G6 (HR 1.66; 95% CI 1.14–2.43)) compared with that of the reference BMI group (G2) (Figure 2).

### 3.7. Association between Body Mass Index, Renal Outcomes, and Mortality According to Hb Levels among Patients with Chronic Kidney Disease Stages 4 and 5

In patients with CKD stages 4 and 5, a high BMI was not associated with an increased HR for renal outcomes (Table 3). To differentiate the effect of BMI and Hb levels in these patients, we reported the HR for renal outcomes and mortality, using the fully adjusted Cox regression model grouped by BMI (Appendix A). In patients with Hb < 11 g/dL, the HR for renal outcomes was significantly increased in patients with a low BMI (G1 (HR 1.41; 95% CI 1.05–1.90)) compared with that of the reference BMI group (G6). In patients with Hb ≥ 11 g/dL, the HR for renal outcomes was significantly increased in the low-BMI (G1 (HR 2.54; 95% CI 1.14–5.63) and G3 (HR 2.12; 95% CI 1.29–3.5)) and high-BMI (G4 (HR 1.97; 95% CI 1.19–3.24), G5 (HR 3.04; 95% CI 1.79–5.18), and G6 (HR 12.64; 95% CI 1.53–4.55)) groups compared with the reference BMI group (G2). 

We also reported the HR for mortality according to BMI, using the fully adjusted Cox regression model (Appendix A). In patients with Hb < 11 g/dL, the HR for mortality was significantly increased in the low-BMI groups (G1 (HR 1.83; 95% CI 1.15–2.90) and G2 (HR 1.57; 95% CI 1.01–2.43)) compared with that of the reference BMI group (G6). In patients with Hb ≥ 11 g/dL, the HR for mortality was not significantly increased in the BMI groups compared with that of the reference BMI group (G5).

### 3.8. Association between the Waist-to-Hip Ratio, Renal Outcomes, and Mortality According to Chronic Kidney Disease Stage

We reported the HR for renal outcomes by using the fully adjusted Cox regression model grouped by WHR (Appendix A). In patients with CKD stages 1–3, the HR for renal outcomes was not significantly increased compared with that of the reference WHR group (Q5). In patients with CKD stages 4 and 5, the HR for renal outcomes was not significantly increased compared with that of the reference WHR group (Q3). 

We also reported the HR for mortality, using the fully adjusted Cox regression model grouped by WHR (Appendix A). In patients with CKD stages 1–3, the HR for mortality was significantly increased in the low WHR group (Q1 (HR 1.82; 95% CI 1.18–2.81)) compared with that of the reference WHR group (Q4). In patients with CKD stages 4 and 5, the HR for mortality was significantly increased in the low-WHR (Q1 (HR 1.41; 95% CI 1.09–1.83)) and high-WHR (Q4 (HR 1.56; 95% CI 1.22–1.99) and Q5 (HR 1.40; 95% CI 1.10–1.78)) groups compared with that of the reference WHR group (Q3).

### 3.9. Association between Waist-to-Hip Ratio, Renal Outcomes, and Mortality According to Hemoglobin Levels

We reported the HR for renal outcomes, using the fully adjusted Cox regression model grouped by WHR (Appendix A). In patients with Hb ≥ 11 g/dL, the HR for renal outcomes was significantly increased in the low-WHR groups (Q1 (HR 1.47; 95% CI 1.05–2.07) and Q2 (HR 1.33; 95% CI 1.00–1.78)) compared with that of the reference WHR group (Q4). In patients with Hb < 11 g/dL, the HR for renal outcomes did not reveal a significant increase in WHR quintiles compared with that of the reference WHR group (Q2). 

We also reported the HR for mortality, using the fully adjusted Cox regression model grouped by WHR (Appendix A). In patients with Hb ≥ 11 g/dL, the HR for mortality did not reveal a significant increase in WHR quintiles compared with the reference WHR group (Q2). In patients with Hb < 11 g/dL, the HR for renal outcomes was significantly increased in low-WHR (Q1 (HR 1.45; 95% CI 1.09–1.92)) and high-WHR (Q4 (HR 1.48; 95% CI 1.12–1.97) and Q5 (HR 1.42; 95% CI 1.08–1.86)) groups compared with that of the reference WHR group (Q3).

## 4. Discussion

The present study included 3605 Asian patients with CKD stages 1–5 and focused on the impact of BMI on renal outcomes in patients at different CKD stages. Our results (Figure 3) revealed that a low BMI was associated with renal function deterioration at all CKD stages, and a high BMI was associated with poor renal outcomes at CKD stages 1–3, rather than stages 4 and 5; this is referred to as the obesity paradox for renal outcomes. In the linear regression model, the BMI was positively associated with Hb and albumin at CKD stages 4 and 5. Our study revealed that a high BMI was associated with poor renal outcomes in patients with Hb ≥ 11 g/dL, but not in those with Hb < 11 g/dL. Central obesity (high WHR) was not associated with renal outcomes and could not explain the obesity paradox for renal outcomes.

Our results demonstrated that both a high and low BMI were related to renal outcomes; however, the effect of a high BMI on patients with CKD stages 4 and 5 was diminished. In patients with CKD stages 1–3, renal function deterioration was statistically significant in both low-BMI (G2 (HR 1.84; 95% CI 1.27–2.66)) and high-BMI (G5 (HR 1.51; 95% CI 1.01–2.26)) groups. In patients with CKD stages 4 and 5, renal function deterioration was only statistically significant in the low-BMI (G1 (HR 1.31; 95% CI 1.07–1.61)) group (Table 3). Studies have reported a U-shaped association between BMI and renal outcomes, suggesting that both a high and low BMI are associated with renal function deterioration in the general CKD population [13,16,27]. However, the effect of BMI was observed to be different at different CKD stages. Our results are consistent with those of Lu et al.’s study, which involved 453,946 veterans in the United States. In patients with earlier stages (eGFR 30–60 mL/min per 1.73 m^2^) of CKD, both a low and high BMI were associated with negative renal outcomes; and in patients with advanced stage (eGFR < 30 mL/min per 1.73 m^2^) CKD, a low BMI (<25 kg/m^2^) was associated with negative outcomes, and the effect of a high BMI (≥35 kg/m^2^) was attenuated [13]. A lower risk of renal function deterioration was observed in patients with advanced CKD as their BMI increased—this is known as the obesity paradox for renal outcomes.

Few studies have investigated the cause of the obesity paradox for renal outcomes. We hypothesized that malnutrition–inflammation and central obesity would modify the association between BMI and renal outcomes at the advanced CKD stage. Our results demonstrated that the effect of a high BMI on renal outcomes was weakened in patients with CKD stages 4 and 5; in our linear regression model, BMI was positively associated with Hb in this population. We further analyzed the association between BMI and renal outcomes in different Hb groups. In the nonanemic (Hb ≥ 11 g/dL) CKD-stage-4/5 population, we observed a statistically significant increase in renal function deterioration in both high-BMI (G3, G5, and G6) and low-BMI (G1) groups; in the anemic (Hb < 11 g/dL) CKD-stage-4/5 population, a low BMI was associated with renal function deterioration, whereas a high BMI was protective. 

The effect of malnutrition–inflammation in patients with advanced CKD and anemia may be the key to explaining the obesity paradox for renal outcomes. Malnutrition–inflammation is associated with PEW [18], which progresses with CKD progression. The inverse relationship between GFR and inflammation is clear [28], and the progression of CKD is closely associated with systemic inflammation and oxidative stress [29]. In a chronic renal insufficiency cohort study, inflammation markers (IL-1β, IL-1 receptor antagonist, IL-6, TNF-α, hs-CRP, and fibrinogen) were associated with decreased kidney function [30]. We reported a reversed J-shaped association of odds ratios for malnutrition–inflammation according to the BMI of patients with CKD stages 3–5 in our previous study [22]. In patients with CKD stages 4 and 5 with a high BMI, malnutrition–inflammation was significantly decreased compared with that of patients with a low BMI, resulting in a protective effect on renal outcomes. In the anemic (Hb < 11 g/dL) CKD-stage-4/5 population, a low BMI was associated with higher renal function deterioration compared with that of the high-BMI population. Anemia is associated with increased mortality, major cardiovascular events, and CKD progression [31], and anemia-related hypoxemia and oxidative stress play a role in tubulointerstitial damage, leading to kidney function deterioration [20]. Anemia had an additional effect on renal function progression in the presence of low-BMI-related malnutrition–inflammation in patients with CKD stages 4 and 5. The malnutrition–inflammation complex is a predictor of poor responsiveness to erythropoiesis-stimulating agents in hemodialysis patients [32,33]. Elevated hepcidin was observed in patients with renal impairment [34] and plays a role in iron metabolism in inflammation-related anemia through the interaction of erythropoietin, iron, and bone marrow [35]. The erythropoiesis-stimulating agent was reimbursed only in patients with CKD stage 5 in our study. A lower level of malnutrition–inflammation anemia in patients with a high BMI and CKD stages 4 and 5 may explain the obesity paradox for renal outcomes, and whether the beneficial effect of adipose tissue is derived from erythropoiesis deserves further study.

Our analysis also revealed that a high WHR was associated with improved—although, not statistically significantly so—renal outcomes in patients with CKD stages 1–3 but was not predictive in those with CKD stages 4 and 5. Obesity results in inflammation, oxidative stress; abnormal lipid metabolism; the activation of the renin angiotensin–aldosterone system; and insulin resistance by producing adiponectin, leptin, and resistin, leading to renal function deterioration [36]. BMI is a standard measurement of obesity but is affected by muscle mass, water retention, fat distribution, and the failure to represent body composition due to individual variations, and some studies have demonstrated that BMI fails to predict the rate of CKD progression [37]. Central obesity is the most prevalent manifestation of metabolic syndrome [38] and is associated with increased cardiovascular disease and mortality [39,40,41]. Increased central fat distribution leads to renal hyperfiltration, hyperperfusion, microalbuminuria, and renal function impairment [42]. Dong et al. conducted a study involving a 29,516 cohort from the general Chinese population and determined that obesity was related to an increased risk of CKD, and percentage body fat was a more effective parameter than other adiposity indices [43]. Pinto-Sietsma et al. concluded that a greater WHR was associated with an increased risk of diminished filtration in 7676 patients without diabetes [42]. In our study, we included Asian CKD participants both with and without diabetes, instead of the general population, including non-CKD patients. Our results revealed that a high WHR was associated with improved renal outcomes, but this was nonsignificant. We further analyzed the effect of Hb levels on renal outcomes grouped by WHR, revealing that a high WHR was related to improved renal outcomes in the non-anemia (Hb ≥ 11 g/dL) group, but no association was identified in the anemia (Hb < 11 g/dL) group. Therefore, we concluded that Hb, but not central fat, explained the obesity paradox for renal outcomes.

We also tested different anthropometric markers for all-cause mortality, revealing that high central obesity is associated with increased mortality, but a high BMI is associated with decreased mortality. BMI is a strong indicator of mortality in the general population, with the relationship between BMI and mortality revealed to be J-shaped in 1.46 million white adults [2] and U-shaped in 0.85 million East Asian adults [3]. The survival advantage of obesity was initially observed by Kalantar et al. in patients undergoing dialysis [6] and recognized as the obesity paradox for mortality. The reverse relationship was also noticed in patients with advanced CKD [44]. Emerging evidence demonstrates that central obesity may be effective at predicting mortality in the advanced CKD population. Postorino et al. [39] and Kim et al. [40] revealed increased all-cause mortality with increasing central obesity in the ESRD population, and Elsayed et al. [45] and Kramer et al. [46] noted increased cardiac events and mortality as central obesity increased in advanced CKD populations. Our previous study identified a reverse J-shaped association between BMI and all-cause mortality [8] and U-shaped association between central obesity and all-cause mortality [22] in the advanced CKD population. The present study revealed that a low BMI was associated with relatively high mortality, and a high BMI was associated with relatively low mortality in patients with CKD stages 1–3 and 4/5. 

Anemia is associated with increased mortality in dialysis-dependent and non-dialysis-dependent patients with CKD [47]. Our study revealed a reverse J-shaped trend between BMI and mortality, which was only statistically significant in low BMI (nonanemic (Hb ≥ 11 g/dL) G2 (HR 2.10; 95% CI 1.19–3.71) and anemic (Hb < 11 g/dL) G1 (HR 1.76; 95% CI 1.16–2.68)) groups; a U-shaped trend was observed between WHR and mortality and was statistically significant in both low-WHR (anemic (Hb < 11 g/dL) Q1 (HR 1.45; 95% CI 1.09–1.92)) and high-WHR (anemic (Hb < 11 g/dL) Q4 (HR 1.48; 95% CI 1.12–1.97) and anemic (Hb < 11 g/dL) Q1 (HR 1.42; 95% CI 1.08–1.86)) groups. The results are consistent with those of other studies indicating an obesity paradox but not a central obesity paradox for mortality in patients with CKD, even with different Hb levels.

The main strength of the present study is the large sample size (*n* = 3605) and enrollment of patients with CKD stages 1–5 with a BMI of 15.0–35.0 kg/m^2^. This study has several limitations. First, baseline anthropometric measurements were used for analysis, and we did not obtain time-dependent changes. Second, our sample was limited to an Asian population, and we could not address racial differences in body composition or outcomes. Third, medication and dietary factors were not included in our study, which may also be crucial to CKD incidence and mortality. Future studies should focus on the obesity paradox for renal outcomes in patients with advanced CKD.

## 5. Conclusions

Our study suggested that CKD stage and anemia may provide an explanation for the obesity paradox for renal outcomes in patients with advanced CKD. A high BMI was associated with poor renal outcomes in nonanemic (Hb ≥ 11 g/dL) patients, as well as those with CKD stages 1–3. The WHR was not associated with renal outcomes. Future studies should consider whether the beneficial effect of adipose tissue is derived from erythropoiesis.

## Figures and Tables

**Figure 1 jcm-11-02787-f001:**
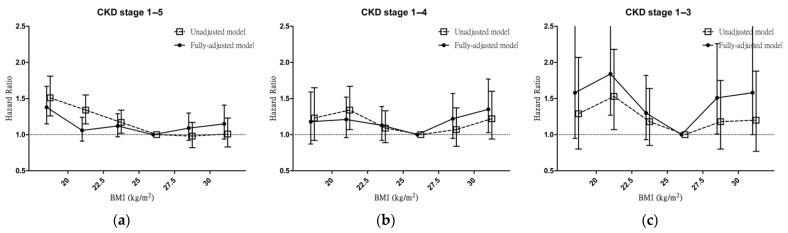
Association between BMI and renal outcomes according to CKD stage. (**a**) Hazard ratios for renal outcomes according to BMI before and after adjustment in patients with CKD stages 1–5. (**b**) Hazard ratios for renal outcomes according to BMI before and after adjustment in patients with CKD stages 1–4. (**c**) Hazard ratios for renal outcomes according to BMI before and after adjustment in patients with CKD stages 1–3. Renal outcomes are defined as renal replacement therapy and a 50% decline in eGFR. Abbreviations: BMI, body mass index; CKD, chronic kidney disease.

**Figure 2 jcm-11-02787-f002:**
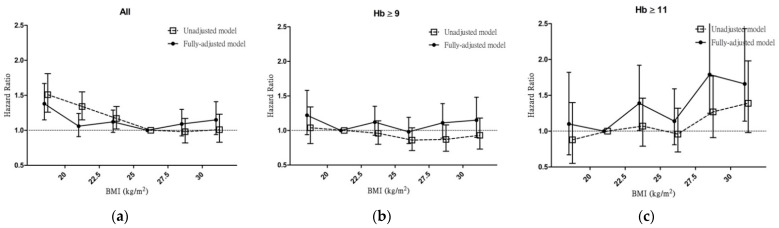
Association between BMI and renal outcomes according to Hb levels. (**a**) Hazard ratios for renal outcomes according to BMI before and after adjustment in all patients. (**b**) Hazard ratios for renal outcomes according to BMI before and after adjustment for patients with Hb ≥ 9. (**c**) Hazard ratios for renal outcomes according to BMI before and after adjustment for patients with Hb ≥ 11. Renal outcomes are defined as renal replacement therapy and a 50% decline in eGFR. Abbreviations: BMI, body mass index; Hb, hemoglobin.

**Figure 3 jcm-11-02787-f003:**
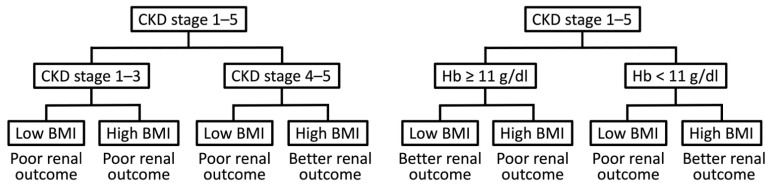
Diagnostic flowchart for renal outcome in CKD patients. Abbreviations: CKD, chronic kidney disease; BMI, body mass index; Hb, hemoglobin.

**Table 1 jcm-11-02787-t001:** Baseline characteristics of body mass index (BMI) according to chronic kidney disease (CKD) stage.

		Body Mass Index (kg/m^2^)					*p*-Value (ANOVA)
	G1	G2	G3	G4	G5	G6	
	15.0–20.0	20.0–22.5	22.5–25.0	25.0–27.5	27.5–30.0	30.0–35.0	
**CKD stages 1–3**							
Demographics/Medical history							
Age (years)	55.3 (20.1)	59.8 (16.2)	63.6 (13.4)	61.6 (13.1)	60.6 (13.8)	58.7 (13.2)	<0.001
Sex (male)	60 (49.6%)	95 (38.8%)	103 (24.6%)	91 (24.1%)	54 (26.0%)	39 (26.9%)	<0.001
Cardiovascular disease	19 (15.7%)	36 (14.7%)	36 (14.7%)	72 (19.1%)	41 (19.7%)	30 (20.7%)	0.574
Diabetes mellitus	45 (37.2%)	99 (40.4%)	185 (44.2%)	177 (46.9%)	103 (49.5%)	87 (60.0%)	0.001
Hypertension	52 (43.0%)	113 (46.1%)	234 (55.8%)	228 (60.5%)	134 (64.4%)	96 (66.2%)	<0.001
Mean BP (mmHg)	93.7 (12.3)	94.9 (13.1)	99.2 (12.8)	100.5 (12.5)	101.8 (12.9)	102.9 (13.2)	<0.001
Antihypertensive drug	37 (30.6%)	86 (35.1%)	155 (37.0%)	174 (46.2%)	112 (53.8%)	64 (44.1%)	<0.001
Charlson score	2.98 (2.04)	3.09 (1.91)	3.19 (1.97)	3.24 (2.03)	3.14 (1.86)	2.81 (1.72)	0.283
Metabolic syndrome	31 (25.6%)	95 (38.8%)	247 (58.9%)	276 (73.2%)	174 (83.7%)	132 (91.0%)	<0.001
Malnutrition–inflammation	36 (29.8%)	23 (9.4%)	25 (6.0%)	23 (6.1%)	12 (5.8%)	10 (6.9%)	<0.001
Laboratory data							
eGFR (mL/min/1.73 m^2^)	44.5 (36.7–70.6)	43.9 (36.4–59.1)	43.5 (36.6–55.7)	45.5 (36.4–57.6)	44.4 (35.9–56.7)	49.0 (38.0–63.2)	0.568
UPCR (mg/g)	626 (168–1735)	389 (170–1529)	421 (166–1361)	425 (145–1201)	377 (164–1174)	493 (171–1322)	0.283
Hemoglobin (g/dL)	11.98 (1.98)	12.39 (1.94)	12.96 (1.88)	13.36 (1.92)	13.46 (1.90)	13.76 (2.08)	<0.001
Albumin (g/dL)	3.90 (0.59)	3.93 (0.54)	3.99 (0.53)	4.00 (0.53)	4.01 (0.49)	3.99 (0.56)	0.272
C-reactive protein (mg/L)	0.5 (0.1–4.7)	0.6 (0.2–3.3)	0.9 (0.3–2.6)	1.0 (0.3–4.1)	0.9 (0.2–5.0)	1.7 (0.5–5.2)	0.757
HbA1c (%)	6.3 (1.8)	6.5 (1.8)	6.5 (1.6)	6.8 (1.8)	6.6 (1.5)	7.1 (1.7)	0.001
Total cholesterol (mg/dL)	202 (58)	205 (65)	203 (54)	204 (69)	198 (48)	203 (63)	0.883
Triglyceride (mg/dL)	101 (54)	132 (92)	156 (207)	159 (102)	176 (107)	177 (104)	<0.001
Outcomes							
RRT + GFR 50% decline	23 (19.0%)	57 (23.3%)	80 (19.1%)	65 (17.2%)	41 (19.7%)	27 (18.6%)	0.039
Mortality	23 (19.0%)	26 (10.6%)	45 (10.7%)	40 (10.6%)	16 (7.7%)	17 (11.7%)	0.047
**CKD stages 4 and 5**							
Demographics/Medical history							
Age (years)	62.53 (15.90)	63.79 (13.45)	63.83 (13.12)	63.56 (13.57)	64.45 (12.43)	61.73 (13.26)	0.353
Sex (male)	147 (61.3%)	226 (54.2%)	281 (46.4%)	199 (45.7%)	120 (50.2%)	94 (61.4%)	<0.001
Cardiovascular disease	67 (27.9%)	102 (24.5%)	170 (28.1%)	128 (29.4%)	80 (33.5%)	51 (33.3%)	0.143
Diabetes mellitus	102 (42.5%)	190 (45.6%)	312 (51.5%)	241 (55.4%)	146 (61.1%)	97 (63.4%)	<0.001
Hypertension	144 (60.0%)	287 (68.8%)	438 (72.3%)	321 (73.8%)	174 (72.8%)	107 (69.9%)	0.004
Mean BP (mmHg)	95.4 (13.9)	99.5 (14.1)	101.0 (13.8)	100.5 (13.3)	102.6 (14.1)	102.2 (15.0)	<0.001
Antihypertensive drug	79 (32.9%)	175 (42.0%)	279 (46.0%)	201 (46.2%)	115 (48.1%)	78 (51.0%)	<0.001
Charlson score	3.38 (2.01)	3.55 (2.25)	3.66 (2.13)	3.74 (2.14)	3.72 (1.91)	3.75 (2.04)	0.281
Metabolic syndrome	103 (42.9%)	235 (56.4%)	421 (69.5%)	373 (85.7%)	216 (90.4%)	143 (93.5%)	<0.001
Malnutrition–inflammation	121 (50.4%)	83 (19.9%)	125 (20.6%)	86 (19.8%)	44 (18.4%)	33 (21.6%)	<0.001
Laboratory data							
eGFR (mL/min/1.73 m^2^)	11.5 (7.2–17.5)	13.0 (8.1–20.3)	13.4 (8.4–21.0)	14.8 (9.3–22.2)	15.0 (9.1–22.8)	15.9 (9.3–22.8)	<0.001
UPCR (mg/g)	1487 (799–3025)	1579 (843–3186)	1612 (792–3477)	1573 (860–3311)	1594 (682–3467)	1538 (679–3709)	0.974
Hemoglobin (g/dL)	9.19 (1.49)	9.47 (1.64)	9.86 (1.81)	10.12 (1.90)	10.31 (1.98)	10.53 (1.80)	<0.001
Albumin (g/dL)	3.67 (0.55)	3.74 (0.55)	3.73 (0.55)	3.77 (0.55)	3.75 (0.51)	3.72 (0.49)	0.362
C-reactive protein (mg/L)	1.1 (0.4–6.9)	1.0 (0.5–5.0)	1.3 (0.5–5.3)	1.5 (0.5–7.4)	2.0 (0.5–7.9)	2.0 (0.5–10.6)	0.002
HbA1c (%)	6.0 (1.4)	6.2 (1.6)	6.4 (1.6)	6.6 (1.7)	6.6 (1.5)	6.9 (1.7)	<0.001
Total cholesterol (mg/dL)	185 (55)	194 (51)	193 (47)	199 (58)	194 (49)	204 (56)	0.005
Triglyceride (mg/dL)	122 (89)	135 (82)	148 (93)	190 (226)	161 (89)	195 (126)	<0.001
Outcomes							
RRT + GFR 50% decline	154 (64.2%)	276 (66.2%)	405 (66.8%)	287 (66.0%)	154 (64.4%)	106 (69.3%)	0.005
Mortality	56 (23.3%)	79 (18.9%)	104 (17.2%)	77 (17.7%)	39 (16.3%)	20 (13.1%)	0.021

Data are presented as mean (standard error), median (interquartile range), or count (percentage %). Abbreviations: CKD, chronic kidney disease; BP, blood pressure; eGFR, estimated glomerular filtration rate; UPCR, urine protein and creatinine ratio; HbA1c, glycosylated hemoglobin; RRT, renal replacement therapy; GFR, glomerular filtration rate.

**Table 2 jcm-11-02787-t002:** Multivariate linear regression for BMI (per 0.05 increase).

	Beta Coefficient	95% CI Beta Coefficient	*p*-Value
Constant	11.749		
Age (years)	0.004	−0.005 to 0.013	0.372
Gender (female vs. male)	0.198	−0.065 to 0.461	0.141
eGFR (mL/min/1.73 m^2^)	−0.005	−0.012 to 0.002	0.132
Upcr log	0.060	−0.199 to 0.319	0.650
Diabetes mellitus	0.726	0.463 to 0.989	<0.001
Cardiovascular disease	0.231	−0.047 to 0.508	0.103
Glycosylated hemoglobin (%)	0.154	0.075 to 0.233	<0.001
Mean BP (mmHg)	0.034	0.026 to 0.043	<0.001
Hemoglobin (g/dL)	0.436	0.364 to 0.509	<0.001
Albumin (g/dL)			
in CKD 1–3	0.007	−0.395 to 0.410	0.972
in CKD 4 and 5	0.427	0.091 to 0.763	0.013
Cholesterol log	0.579	−0.513 to 1.670	0.299
CRP ln	0.340	0.208 to 0.471	<0.001
Phosphorus (mg/dL)	−0.041	−0.162 to 0.079	0.498
Uric acid (mg/dl	0.211	0.149 to 0.272	<0.001
Ferritin log	−0.235	−0.493 to 0.023	0.074
GPT (U/L)	0.004	−0.001 to 0.009	0.096
Bicarbonate (mEq/L)	−0.037	−0.070 to −0.004	0.027

Abbreviations: CI, confidence interval; eGFR, estimated glomerular filtration rate; Upcr, urine protein and creatinine ratio; BP, blood pressure; CKD, chronic kidney disease; CRP, C-reactive protein; GPT, glutamic oxaloacetic transaminase.

**Table 3 jcm-11-02787-t003:** Association between BMI, renal outcomes, and mortality according to CKD stage.

		BMI (kg/m^2^)					
		G1	G2	G3	G4	G5	G6
		15.0–20.0	20.0–22.5	22.5–25.0	25.0–27.5	27.5–30.0	30.0–35.0
HR for renal outcome							
CKD stages 1–3	Unadjusted	1.29 (0.80–2.07)	1.53 (1.07–2.18) *	1.18 (0.85–1.64)	1 (reference)	1.18 (0.80–1.75)	1.20 (0.77–1.88)
	Fully adjusted	1.58 (0.95–2.61)	1.84 (1.27–2.66) *	1.30 (0.93–1.82)	1 (reference)	1.51 (1.01–2.26) *	1.58 (1.00–2.51)
CKD stages 4 and 5	Unadjusted	1.26 (1.03–1.53) *	1.06 (0.90–1.25)	1.02 (0.88–1.19)	1 (reference)	0.90 (0.74–1.10)	0.91 (0.73–1.14)
	Fully adjusted	1.31 (1.07–1.61) *	0.96 (0.81–1.13)	1.05 (0.90–1.22)	1 (reference)	1.02 (0.84–1.25)	1.04 (0.83–1.31)
HR for all-cause mortality							
CKD stages 1–3	Unadjusted	2.05 (1.28–3.26) *	1.33 (0.87–2.03)	1.21 (0.83–1.77)	1 (reference)	0.86 (0.52–1.43)	1.22 (0.73–2.03)
	Fully adjusted	2.54 (1.55–4.16) **	1.37 (0.88–2.14)	1.23 (0.84–1.81)	1 (reference)	1.01 (0.60–1.68)	1.44 (0.86–2.42)
CKD stages 4 and 5	Unadjusted	1.64 (1.14–2.37) *	1.45 (1.03–2.05) *	1.33 (0.95–1.85)	1.27 (0.90–1.80)	1.24 (0.85–1.80)	1 (reference)
	Fully adjusted	1.91 (1.30–2.80) **	1.58 (1.11–2.25) *	1.33 (0.94–1.86)	1.27 (0.89–1.80)	1.18 (0.81–1.72)	1 (reference)

Values expressed as hazard ratio (HR) and 95% confidence interval (CI). Fully adjusted model: adjusted for age, sex, eGFR, Upcr log, cardiovascular disease, smoking history, cancer, severe liver disease, and hypertension; * *p* < 0.05 compared with reference BMI category; ** *p* < 0.001 compared with reference BMI category. Renal outcomes are defined as renal replacement therapy and a 50% decline in eGFR. Abbreviations: BMI, body mass index; CKD, chronic kidney disease; HR, hazard ratio; Upcr, urine protein and creatinine ratio.

**Table 4 jcm-11-02787-t004:** Association between BMI, renal outcomes, and mortality according to hemoglobin (Hb) levels.

		BMI (kg/m^2^)					
		G1	G2	G3	G4	G5	G6
		15.0–20.0	20.0–22.5	22.5–25.0	25.0–27.5	27.5–30.0	30.0–35.0
HR for renal outcome							
Hb < 11 g/dL	Unadjusted	1.19 (0.93–1.52)	1.07 (0.86–1.34)	1.13 (0.91–1.39)	1.15 (0.92–1.44)	1 (reference)	1.05 (0.78–1.41)
	Fully adjusted	1.48 (1.15–1.90) *	1.15 (0.92–1.45)	1.15 (0.93–1.43)	1.09 (0.87–1.38)	1 (reference)	1.03 (0.76–1.39)
Hb ≥ 11 g/dL	Unadjusted	0.88 (0.55–1.40)	1 (reference)	1.07 (0.79–1.46)	0.96 (0.71–1.32)	1.27 (0.91–1.77)	1.39 (0.98–1.98)
	Fully adjusted	1.10 (0.67–1.82)	1 (reference)	1.39 (1.01–1.92) *	1.14 (0.81–1.59)	1.79 (1.25–2.55) *	1.66 (1.14–2.43) *
HR for mortality							
Hb < 11 g/dL	Unadjusted	1.51 (1.00–2.27) *	1.27 (0.86–1.88)	1.21 (0.82–1.78)	1.13 (0.75–1.69)	1.19 (0.77–1.85)	1 (reference)
	Fully adjusted	1.76 (1.16–2.68) *	1.41 (0.95–2.11)	1.15 (0.77–1.70)	1.09 (0.72–1.64)	1.02 (0.65–1.59)	1 (reference)
Hb ≥ 11 g/dL	Unadjusted	0.98 (0.65–1.48)	1.23 (0.72–2.12)	1.08 (0.69–1.69)	1 (reference)	1.09 (0.73–1.62)	0.88 (0.56–1.40)
	Fully adjusted	1.12 (0.73–1.70)	2.10 (1.19–3.71) *	1.48 (0.92–2.36)	1 (reference)	1.30 (0.86–1.98)	1.08 (0.67–1.73)

Values expressed as hazard ratio (HR) and 95% confidence interval (CI). Fully adjusted model: adjusted for age, sex, eGFR, Upcr log, cardiovascular disease, smoking history, cancer, severe liver disease, and hypertension; * *p* < 0.05 compared with reference BMI category. Renal outcomes are defined as renal replacement therapy and a 50% decline in eGFR. Abbreviations: BMI, body mass index; Hb, hemoglobin; HR, hazard ratio; Upcr, urine protein and creatinine ratio.

## Data Availability

Study data are available from the corresponding author (F.-C.S.) upon request.

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
