# Peer review of "Association between Body Mass Index and Renal Outcomes Modified by Chronic Kidney Disease and Anemia: The Obesity Paradox for Renal Outcomes"

_jcm, 2022, doi:10.3390/jcm11102787_

Round 1
Reviewer 1 Report
In the present study, the authors examined the association of body mass index (BMI) and renal outcomes. BMI was positively associated with hemoglobin and albumin at CKD stage 4–5. A high BMI was associated with renal outcomes at CKD stage 1–3 but not 4–5. A high BMI was also associated with renal outcomes in patients with hemoglobin ≥11 g/dl. A high waist-to-hip ratio was not associated with renal outcomes. The authors concluded that the CKD stage and anemia may explain the obesity paradox in renal outcomes in patients with CKD.
The findings are interest and I have several concerns.
- Several studies previously presented the association of BMI and renal outcomes CKD Care Program. Authors should clearly describe the positioning and significance of the present study. In addition, it would better to add several comments of time-course changes including eGFR and anemia.
- The authors examined the urine protein and creatinine ratio from a random spot urine sample. Authors should add the data of urine protein in the Tables.
- Please added a detailed description of msTaiwan3, MIS ≥ 8, and Charlson score.
- In the Table 1, did the difference of ratio of diabetes and hypertension between groups influence the renal outcome in the present study? It would be better to add the information of average score of HbA1c, BP and medications.
- It should be carefully interpreted the obesity paradox including basal characteristics and a causal relationship. At CKD stage 4–5, severe renal dysfunction itself induced malnutrition, catabolism and weight loss. As a result, low BMI, but not high BMI, was associated with renal dysfunction at CKD stage 4–5?
- A high BMI was associated with renal outcomes in patients with hemoglobin ≥11 g/dL. A high waist-to-hip ratio was not associated with renal outcomes. It is interesting points and authors should discuss in greater detail.
Reviewer 2 Report
Review for the manuscript entitled “Association Between Body Mass Index and Renal Outcomes Modified by Chronic Kidney Disease and Anemia: The Obesity Paradox for Renal Outcomes”.
This study examines the impact of obesity on renal outcomes and death. Anemia is employed as the main confounding factor, but several other abdominal obesity indices and clinical parameters are also analyzed. In conclusion, low BMI and anemia could contribute to renal outcomes. On the other hand, the Tables are very complex, and it is therefore difficult to understand how the results of this study explain the obesity paradox.
Concerns:
- Why did the authors consider that the effect of anemia on outcomes varied with body mass index? The reason does not seem to have been mentioned in the introduction.
- In Table 2 (Multivariate regression model for BMI), it seems undesirable to employ several factors that are clearly intra-correlated at the same time. In the first place, why analyze BMI, one of the explanatory variables (and a factor that is not linearly related to the outcome), instead of examining the factors involved in the outcome? Normally, one would expect to make group comparisons by outcome or to present correlations for outcomes.
- Kaplan-Meier curves separated by BMI stage should be presented for each of the two outcomes.
4, The rationale for selecting the confounding factors used in the Cox-hazard proportional analyses is unclear.
- Table 4: The effect of body mass index, tested separately for anemia status, seems to be completely different for the two outcomes. Why?
- Is the obesity paradox recognized in the present study? Obese people appear to be associated with increased outcomes.
- In the end, which is more useful for predicting renal outcomes, BMI or WHR?
- Significant and non-significant HRs are mixed, as multiple outcomes, anemia status, CKD stage and obesity grade provide a vast matrix. As a result, consistent results are difficult to accept, and the authors' discussion is complex. While precision is required in scientific papers, it is also important to be concise so that the reader can easily translate the results into clinical practice.
If the significance of body adiposity indices with anemia or CKD stage, it may be worthwhile to create a flowchart that divides the cases in that way.
That's all.
Round 2
Reviewer 1 Report
The revision has improved the manuscript. I have no further concern.
Reviewer 2 Report
The authors were appropriately addressed in response to the initial review comments.
This manuscript is a resubmission of an earlier submission. The following is a list of the peer review reports and author responses from that submission.